# Metal Coating Synthesized by Inkjet Printing and Intense Pulsed-Light Sintering

**DOI:** 10.3390/ma12081289

**Published:** 2019-04-19

**Authors:** Fanbo Meng, Jin Huang, Haitao Zhang, Pengbing Zhao, Peng Li, Chao Wang

**Affiliations:** Mechanical and Electrical Engineering, Xidian University, Xi’an 710071, China; htzhang1223@163.com (H.Z.); pbzhao@xidian.edu.cn (P.Z.); yinhong0523@163.com (P.L.); 15091620500@163.com (C.W.)

**Keywords:** inkjet printing, metal coating, flash sintering, adhesion

## Abstract

The inkjet printing of nanoparticle inks to produce metal coatings is low in manufacturing cost and high in efficiency compared to conventional methods such as electroplating and etching. However, inkjet-printed metal coatings require sintering to provide better metal conductivity and adhesion. Traditional sintering methods require high processing temperatures that can easily damage the coating substrate. In this study, an enhanced overall conductivity is achieved by sintering a nanoparticle metal coating with intense pulsed light. Metal coatings sintered using different parameters were characterized by a profilometer and a four-probe tester, which showed that the surface topographies differed with different sintering degrees. The adhesion of the metal coating was proportional to the pre-sintering temperature within the allowable range of the substrate. Finally, the optimization of the sintering process according to the experimental results improved both the electrical conductivity and adhesion of the metal coating. The optimized parameters were used to fabricate a microstrip antenna and perform the return loss test and microwave darkroom test. The results matched the simulation results well.

## 1. Introduction

As an additive manufacturing process, compared to traditional manufacturing methods, inkjet printing has the advantages of environmental friendliness, low-cost, simple processing, and suitability for complex patterns by depositing conductive inks on flexible substrates, as controlled by computer graphics data [1]. Inkjet printing technology has been explored as a novel method for printing mammalian cells [2], organic light-emitting diodes [3], solar cells [4], and electronic devices [5]. The printed electronics technique has wide potential applicability in flexible electronic devices [6] and radio-frequency devices, such as wide-band high-gain antennas fabricated by inkjet printing on low-cost substrates [7] and high power microwave amplifiers printed on alumina [8]. The most commonly used materials in printed electronics are nanoscale metal inks, including gold [9], silver [10], and copper [11]. Despite its excellent properties, nanoscale gold ink is limited in applicability by its high price. Copper nanoparticles are easily oxidized under ambient conditions. However, nanoscale silver ink has attracted much attention because of its stable properties and relatively low price.

Printed patterns must be well sintered to obtain excellent electrical conductivity and surface quality [12]. The melting point of the metal in the ink is obviously decreased by the nanosize effect [13]. Temperature is an important parameter in sintering. Generally, in the appropriate range, higher temperatures correspond to better conductivity. During traditional thermal sintering in ovens, the substrate and the printed coating are both heated. The most common substrates are polymer-based with low glass transition temperatures and poor thermal stabilities [14] and are therefore, easily deformed. For laser sintering [15], when sintering a printed gold electrode with a continuous-wave argon ion laser, Chung et al. found that the ink was displaced ahead and around the scanning laser spot, forming a U-shaped convex ink meniscus due to the Marangoni effect [16]. In addition, laser sintering scan speeds range from 50 mm/s to 100 mm/s, and it takes 16–33 min to scan a square with a side length of 100 mm, which is relatively time consuming and inefficient. 

Recent research has reported that intense pulsed-light sintering is an ideal method with the advantages of high speed, high efficiency, and room-temperature sintering of metal materials [17,18,19,20]. The broad-wavelength intense pulse light emitted by a xenon lamp irradiated on metal film interacts with the metal nanoparticles to form sintered necks. These sintered necks grow to form a continuous conductive region. Kang et al. calculated the increase in temperature of a 10 nm thick silver nanoparticle ink pattern of 500 °C during flash sintering, based on the lumped capacitance model [21], which was sufficient to fully melt the silver nanoparticles. At present, most investigations on intense pulsed-light sintering have relied on experiments, wherein the sintering parameters are difficult to control and defects in the conductive pattern quality are easily caused. Lee et al. found that consecutive light pulses from a xenon lamp induced film swelling and even local film ruptures, with corresponding hollow microstructures of the inkjet-printed silver nanoparticle films [22].

In this study, an integrated inkjet printing device with an intense pulsed-light sintering module consisting of a high-power xenon lamp, flash controller, and piezoelectric inkjet printing module, was developed to study the intense pulsed-light sintering process compared to numerical simulation. By optimizing the processing parameters, a patch antenna with excellent conductivity and surface quality was successfully printed and sintered using silver nanoparticle-based conductive ink on an epoxy resin substrate. The return loss and the pattern of antennas were tested in a microwave darkroom. The results showed that the patch antenna made by inkjet printing and flash sintering achieved performance comparable to that of the simulation model. In addition, the surface roughness of the conductive patterns formed under different sintering process parameters was measured using a surface gauge. The results showed a correlation between the surface roughness and electrical conductivity, which provided a new idea for judging the quality of flash intense pulsed-light sintering.

## 2. Materials and Methods 

In order to realize the inkjet printing and intense pulsed-light sintering of a metal coating, an integrated printing and sintering device was developed, as shown in Figure 1, which included a control software system, control hardware system, XYZ motion axis (Chengdu Fuyu Technology Co., Ltd., Chengdu, China), piezoelectric nozzle (Micro-Fab Technologies, Inc., Plano, TX, USA), constant-temperature heater, and flash lamp. The metal coating software model was converted into the STL format and imported into the control software system. The software system analyzed the model data, sliced it according to the printing parameters, and generated the print commands, which were sent to the control hardware system. The hardware system obtained the printing coordinates, print interval, acceleration, deceleration, and other printing parameters, started the XYZ motion axis to execute the print command, and controlled the piezoelectric nozzle to spray the silver nanoparticle ink at the specified intervals. During the printing, the constant-temperature heater ensured that the metal-coated substrate was held at 55 °C. After printing, the flash lamp was activated, and the conductive ink was sintered to obtain a metal coating.

The silver nanoparticle conductive ink (Jet-100, HS Electronics Inc, Suzhou, China) was obtained commercially and mainly comprises silver nanoparticles, organic resin, and a mixture of ethanol and small-molecule alcohol ethers with medium and high boiling points. The sintering process was mainly divided into the pre-sintering and formal sintering stages, as shown in Figure 2. In the pre-sintering stage, the conductive ink was brought to 50–60 °C and the low boiling point solvents such as ethanol are volatilized, forming a flat semi-dried silver nanoparticle film. At this time, the silver nanoparticles were not connected and the film was non-conductive. In the formal sintering stage, the flash lamp was turned on. At this time, the medium and high boiling point solvents began to volatilize and the concentration of nanoparticles in the silver film increased, but the silver particles remained unattached and the film remained non-conductive. As the intense pulsed-light sintering continued, a large amount of energy was transferred in a short time to sinter the silver nanoparticles, which became connected as a continuous sintered phase that formed a silver metal film surface with conductive properties.

The piezoelectric nozzle had a diameter of 60 μm, a drive waveform of trapezoidal waves, and a maximum injection frequency of 15 kHz. The drop diameter was between 45 and 55 μm. The maximum spray viscosity was 15 cP. The flash lamp had a power of 4 kW supplied by a pulse power source, a wavelength range of 230–680 nm, and a sintering time of 1–1.5 ms. The flash sintering power density could be arbitrarily adjusted in the range of 10–25 J/cm^2^. 

## 3. Results and Discussion

### 3.1. Intense Pulsed-Light Sintering

A metal coating was obtained by intense pulsed-light sintering of the silver nanoparticle printed layer. Pulsed-light sintering is characterized by a short time and low sintering process temperature; however, the flash lamp releases significant energy in short periods. Because the sintering effect is unstable [23,24], we observed by a microscope that after experiencing inappropriate flash sintering, the surface of the conductive pattern was broken and the electrical conductivity was lowered. We thus needed to obtain a lower-resistivity metal coating by adjusting the sintering parameters. In order to optimize the appropriate sintering parameters, different sintering times and power densities were tested; the conductivities of the nano-silver metal coatings were measured using a dual-electric four-probe tester (RTS-9). Figure 3a shows the change in flash sintering energy density from 13.6 J/cm^2^ to 23.4 J/cm^2^. In applying one to eight intense pulses at each energy density, the optimum resistivity of 15.7 μΩ/cm was obtained at an energy density of 13.6 J/cm^2^. This high resistivity value was because the low flash power could not completely drive the melting of the silver nanoparticles to form sintered necks, and the silver nanoparticles remained independent of each other. At an energy density of 17.9 J/cm^2^ and three applied pulses, the lowest resistivity was 13.1 μΩ/cm. However, with increasing numbers of pulses or increasing energy densities, the resistivity began to increase, as shown in Figure 3a. The lowest resistivities obtained at the energy densities of 20.3 J/cm^2^ and 23.4 J/cm^2^ were 14.1 μΩ/cm and 15 μΩ/cm, respectively, both of which exceeded 13.1 μΩ/cm. Excessively high energy densities or excessive pulses destroyed the sintered necks formed between the silver nanoparticles, which decreased the electrical conductivity. 

The temperature of the sintered coating was monitored using a thermal imager (Ti300, Fluke, Everett, WA, USA). It can be seen from Figure 3b that the maximum temperature after eight sintering pulses was 52.6 °C. Therefore, many non-high-temperature-resistant materials could be used as substrates.

### 3.2. Pre-Sintering Parameter Analysis

The nano-silver metal coating required pre-sintering before it was fully sintered in order to evaporate most of the low boiling point solvent. The pre-sintering temperature in the experiment significantly affected the adhesion and electrical resistivity of the conductive coating. In order to reveal this mechanism of influence, pre-sintering contrast experiments at different temperatures were performed, and the sample resistivity and adhesion were tested using a dual-electric four-probe tester (RTS-9, Hangzhou Tibetan Han Technology Co., Ltd., Hangzhou, China) and cross-cut testing. The experimental results are shown in Figure 4. Figure 4a shows the conductive coating obtained after pre-sintering at 40 °C. A bulging bulb could be seen in the enlarged area; the low pre-sintering temperature did not completely volatilize the solvent. When full sintering was performed, the solvent at the bottom was volatilized upward by the intense pulsed-light energy; this destroyed the surface of the formed metal coating, leaving a mark from the bulge. The voids generated by the evaporation of the solvent not only reduced the electrical conductivity of the metal coating, but also reduced the adhesion of the metal coating to the substrate, as shown in Figure 4d, with the adhesion grade of only 3 B. When the pre-sintering temperature was adjusted to 60 °C, as shown in Figure 4b, the surface morphology was uniform, the resistivity drops to 13.1 μΩ/cm, and the adhesion was the highest level of 5B. As shown in Figure 4c, the pre-sintering temperature was 80 °C. It can be seen from the magnified section that the high temperature caused surface oxidation during pre-sintering. The oxidized surface hindered subsequent sintering and reduced the sintering degree, although the adhesion grade was still 5B. In addition, the resistivity increased to 15.6 μΩ/cm.

### 3.3. Analysis of Metal Coating Surface Morphology

After the pre-sintering of the silver nanoparticle ink, the ink solvent was volatilized and the silver nanoparticles were unevenly deposited on the surface of the substrate. After sintering, the silver nanoparticles were connected, forming a relatively uniform metal coating due to the Marangoni effect. During this period, the surface topography of the metal coating evolved from pre-sintering unevenness to post-sintering flatness. As shown in Figure 5, we used a surface gauge (TR200, Shenzhen Junda Instrument Co., Ltd., Shenzhou, China) to measure the roughness of silver metal coatings processed with different sintering times. For the coating after intense pulsed-light sintering, the surface morphology was as shown in Figure 5a with a roughness R_a_ = 0.597 μm. After three sintering pulses, the roughness R_a_ decreased to 0.392 μm. At this time, the surface morphology of the metal coating was suitably flat and the overall electrical resistivity was low. For eight sintering pulses, as shown in Figure 5d, the excessively high deposited energy caused the breakage of the conductive pattern surface and the surface roughness increased to 0.735 μm.

To further prove this rule, we divided the metal coating into nine regions, as shown in Figure 6. After one to eight sintering pulses, each region was measured using a profiler to calculate the roughness. The roughness of the nine regions progressed from high to low, to high, as the number of sintering pulses increased. Thus, the surface topography and roughness could be used to determine the degree of sintering in future research, allowing monitoring and optimization of the sintering process to promote the improvement of sintering quality.

## 4. Microstrip Antenna Fabrication and Testing

In this study, inkjet printing and flash sintering were used to fabricate array microstrip antennas. The dimensions of the microstrip antenna are shown in Figure 7a. The parameters of the dielectric substrate are shown in Table 1. The dielectric constant was 4.4 and the dielectric loss was 0.02. The antenna design used a microstrip line antenna feed. We used the simulation software HFSS for the antenna optimization simulation. After the optimization, the size of the array was 13.4 mm × 21.7 mm. The droplet print pitch was controlled to 0.1 mm. An SMA (Sub-Miniature-A) connector was soldered to the wave port to realize a microstrip antenna.

In order to determine the relationship between the sintering parameters and the radiation performance of the microstrip antenna, we fabricated several array microstrip antennas for different experiments using different sintering powers, as shown in Figure 7b, using a vector network analyzer (R&S ZNC, Rohde & Schwarz, Munich, Germany) for return loss testing. As shown in Figure 7c, the sintered power densities of 12 J/cm^2^ and 24 J/cm^2^ lead to incomplete and excessive sintering, respectively, which both increased the resistivity and deteriorated the return loss. At the power density of 18 J/cm^2^, the return loss result was the best, and the result was compared with the simulation result. As shown in Figure 7d, the design center frequency error was within 0.1 GHz and the −15 dB bandwidth of the antenna was 0.42 GHz. The pattern test of the inkjet print array microstrip antenna was performed using a microwave darkroom. Compared with the simulation results, as shown in Figure 7e,f, the main lobes of the E- and H-plane patterns were consistent with the trends shown by the simulation results. This also proved the importance of optimization of sintering parameters.

## 5. Conclusions

In this study, intense pulsed light was used to sinter inkjet-printed silver nanoparticle ink to obtain a conductive metal coating. The resistivity decreased to 13.1 μΩ/cm by adjusting the sintering parameters and the mechanism by which the pre-sintering parameters affected the properties of the metal coating was explained. The relationship between the sintering parameters and sintering degree was established, and the sintering parameters were adjusted and optimized to improve the electrical performance. This optimized process was applied to the fabrication of array microstrip antennas. The relationship among sintering degree, resistivity, and antenna radiation were verified by comparison of the return loss for antennas prepared with different sintering powers. The array antenna pattern results were consistent with the trends of the simulation results. This study relates the surface morphology of the metal coating to the degree of sintering. The surface morphology thereby allows determination of the sintering state, which may be used for the subsequent realization of a closed-loop sintering process. Our findings could effectively improve the efficiency of the sintering process in the electronic printing method, and provide a basis for manufacturing flexible circuits, RF antennas, and small sensors for low-cost and high speed.

## Figures and Tables

**Figure 1 materials-12-01289-f001:**
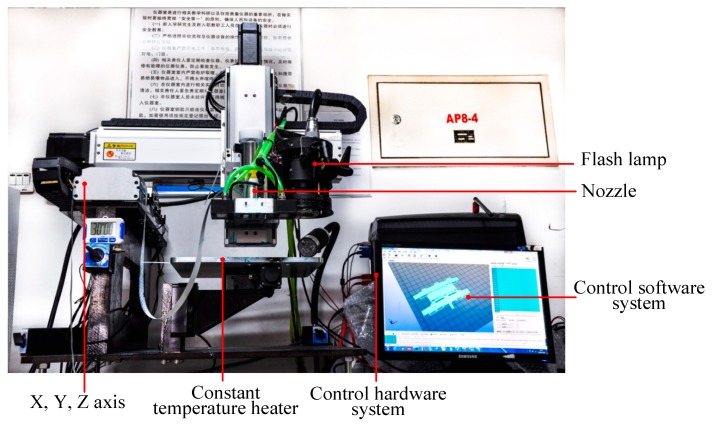
Schematic of inkjet-printing and sintering metal coating equipment.

**Figure 2 materials-12-01289-f002:**
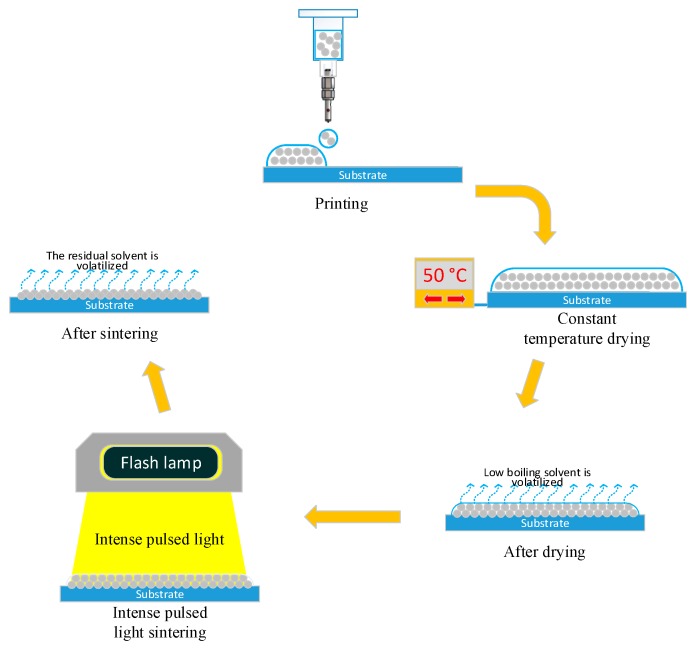
Schematic of the inkjet-printing metal coating process.

**Figure 3 materials-12-01289-f003:**
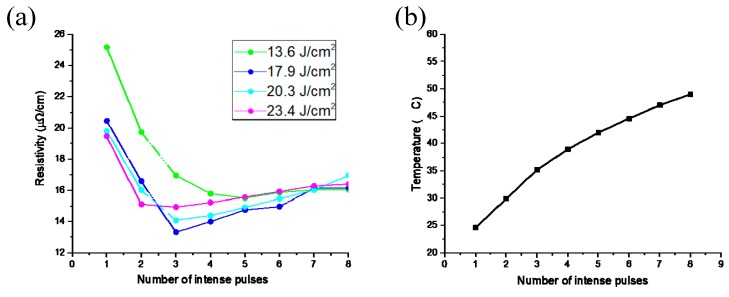
(**a**) Relationship between sintering times and resistivity at different sintering power densities (**b**) Relationship between sintering times and temperature at a power density of 17.9 J/cm^2^.

**Figure 4 materials-12-01289-f004:**
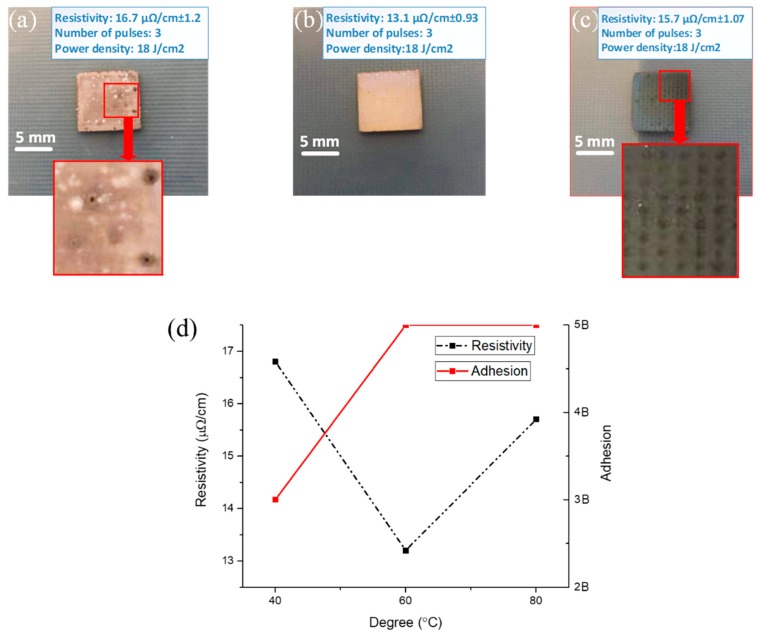
Samples with pre-sintering temperatures of (**a**) 40 °C, (**b**) 60 °C, and (**c**) 80 °C. (**d**) Resistivities and adhesion grades of the samples with different pre-sintering temperatures.

**Figure 5 materials-12-01289-f005:**
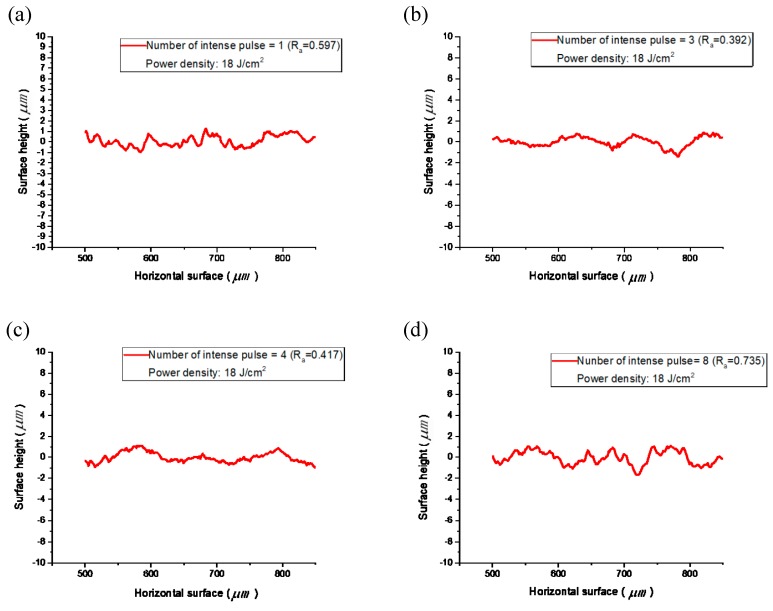
Surface profiles of samples after (**a**) one, (**b**) three, (**c**) four, and (**d**) eight sintering pulses.

**Figure 6 materials-12-01289-f006:**
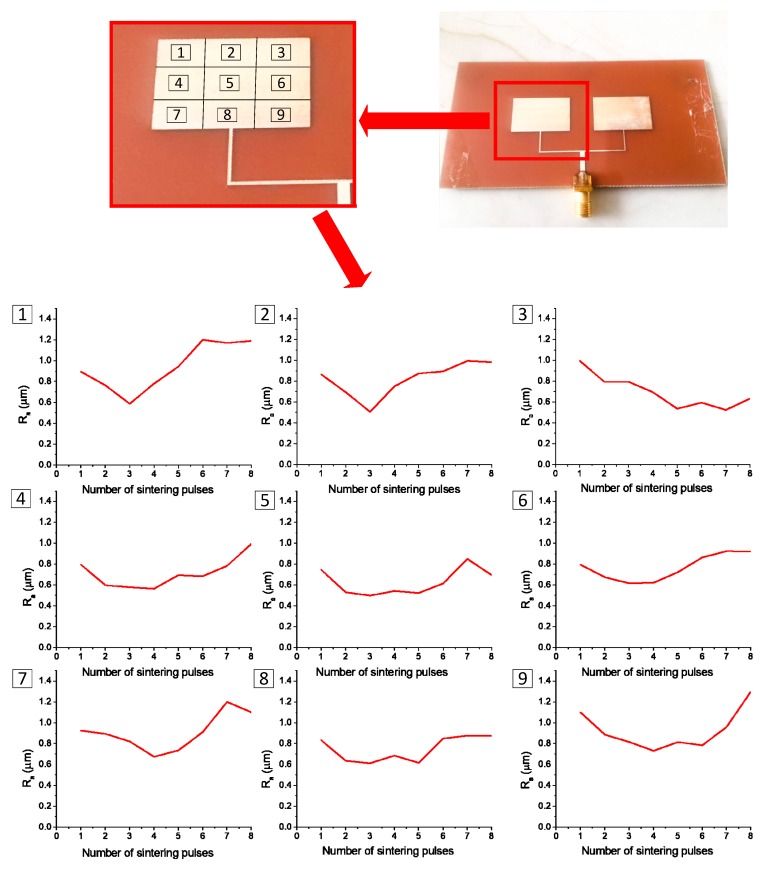
Surface topographies of nine regions of a square sample, as tested using a profiler after sintering pulses; plots show relationships between the number of sintering pulses and the surface roughness for each region.

**Figure 7 materials-12-01289-f007:**
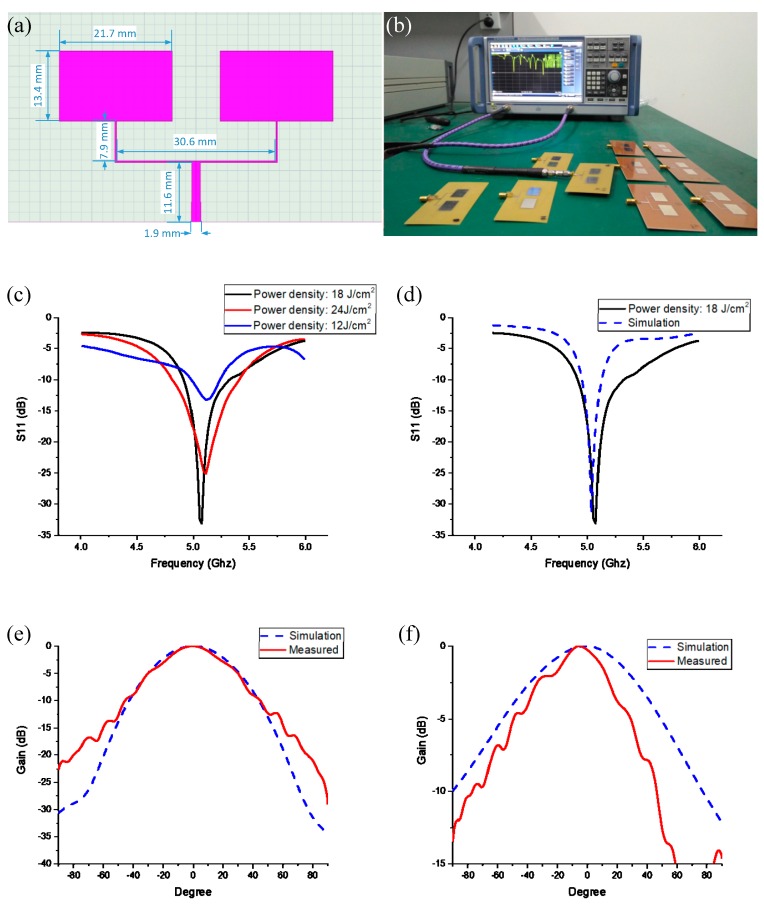
(**a**) Array microstrip antenna size diagram. (**b**) Array microstrip antenna test S11 data. (**c**) Microstrip antenna S11 data of different sintering power densities. (**d**) Comparison of S11 data between microstrip antenna with sintered power density of 18 J/cm^2^ and simulation results. (**e**) Comparison of E-plane pattern data and simulation results of microstrip antenna with sintered power density of 18 J/cm^2^. (**f**) Comparison of H-plane pattern data and simulation results of microstrip antenna with sintered power density of 18 J/cm^2^.

**Table 1 materials-12-01289-t001:** Substrate information.

Permittivity	Dielectric Loss Angle (°)	Thickness (mm)
4.4	0.02	1

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
