# Peer review of "Metal Coating Synthesized by Inkjet Printing and Intense Pulsed-Light Sintering"

_materials, 2019, doi:10.3390/ma12081289_

Reviewer 1 Report

Overall the work is interesting and may add to a field that is growing fast.

However there are several points to be considered to potentially increase the quality of your output.

Abstract

Line 11 "Traditional"

Introduction

Page 1

Line 31 add also high power microwave amplifiers printed on alumina [Journal of Alloys and Compounds 615 (2014) S501–S504]

Line 32 after silver add a relevant review [Nanotechnology, Science and Applications 2016:9 1–13]

Line 36/37 reference missing, find one suitable

Line 37 reference missing, find one suitable

Line 42 "continuous wave"

Page 2

Line 44 another work achieved interesting %bulk of silver [Microelectronic Engineering 88 (2011) 2481–2483]

"laser sintering is relatively time-consuming" you should be more specific and put numbers, eventually create a new table confronting available technologies

Materials & Methods

Line 86 company name?

Line 87 "mixture" instead of "solvent"

Line 90 "ink is brought to"

Line 95 "energy is transferred"

Intense pulsed-light sintering

Line 105 "printed layer" instead of "coating"

Line 107 what do you mean with instable layer? The sintering by itself can be stable!

Figure 4 add on top of the panels the sintering energy and number of pulses or "prior to sintering" if the case; panel d) show only dots, when you have just a few experimental points.

Figure 5 (b) is it possible to show the curve for 2 pulses?

Figure 6 all graphics supanels miss error bar

Please complete chapter 4, it's very short and too superficially discussed!

Author Response

1.Question.

Abstract

Line 11 "Traditional"

Introduction

Page 1

Line 31 add also high power microwave amplifiers printed on alumina [Journal of Alloys and Compounds 615 (2014) S501–S504]

Line 32 after silver add a relevant review [Nanotechnology, Science and Applications 2016:9 1–13]

Line 36/37 reference missing, find one suitable

Line 37 reference missing, find one suitable

Line 42 "continuous wave"

Page 2

Line 44 another work achieved interesting %bulk of silver [Microelectronic Engineering 88 (2011) 2481–2483]

"laser sintering is relatively time-consuming" you should be more specific and put numbers, eventually create a new table confronting available technologies

Materials & Methods

Line 86 company name?

Line 87 "mixture" instead of "solvent"

Line 90 "ink is brought to"

Line 95 "energy is transferred"

Intense pulsed-light sintering

Line 105 "printed layer" instead of "coating"

Answer.

Thank you for your review, I have modified the error and added some references. The modified part is highlighted in red in the manuscript.

 2. Question

Line 107 what do you mean with instable layer? The sintering by itself can be stable!

Figure 4 add on top of the panels the sintering energy and number of pulses or "prior to sintering" if the case; panel d) show only dots, when you have just a few experimental points.

Answer

Thank you for your review, I have added references and explained them in the manuscript.

I have modified the figure as required.

The modified part is highlighted in red in the manuscript.

3. Question

Figure 5 (b) is it possible to show the curve for 2 pulses?

Answer

Thank you for your review. The reason why it is not shown in the manuscript is because the result is close to one pulse and the gap cannot be seen.

The curve for 2 pulses result figure is as follows:

4. Question

Figure 6 all graphics supanels miss error bar

Answer

Thank you for your review. In this picture, the conductive pattern is divided into 9 parts, and the roughness measurement is performed under different sintering times. The results show that the roughness variation trends of the nine sections are the same, starting from high to low and then high. trend. The purpose of dividing the conductive pattern into nine parts is to prevent experimental contingency and improve the credibility of the result. So there is no error added.

5. Question

Please complete chapter 4, it's very short and too superficially discussed!

Answer

Thank you for your review. I have added some detailed information to the manuscript.

The modified part is highlighted in red in the manuscript.

Reviewer 2 Report

The paper is well organized, understandable and sound.

However, results shown in chapter 3.3 should be presented in a better visibility Fig. 5 does not really show the differences in roughness discussed. This part needs improvement in order to become comprehensible.

Some words  on applicability of this technique in practice shall be added to the conclusions.

Author Response

1. Question.

However, results shown in chapter 3.3 should be presented in a better visibility Fig. 5 does not really show the differences in roughness discussed. This part needs improvement in order to become comprehensible.

Answer.

Thank you for your review. I have added the sintered power density to the figure and thickened the lines to give the reader a clearer view of the roughness variation at different sintering times.

2. Question.

Some words on applicability of this technique in practice shall be added to the conclusions.

Answer.

Thank you for your review. I have added potential application areas in the conclusion section.

The modified part is highlighted in red in the manuscript.

Reviewer 3 Report

1) Melting point of a metal and the size of specimen.

2) Explain the Marangoni Effect.

3) Explain the figures 3a.

4) Adhesion and pre-sintering temperature.

5) Explain the figure 4.

Author Response

1. Question.

Melting point of a metal and the size of specimen.

Answer.

Thank you for your review. The melting point of a metal is is 130 degrees. The size of the specimen is 10mm*10mm.

2. Question

Explain the Marangoni Effect.

Answer

Thank you for your review. The reason for the Marangoni effect is that the liquid with a large surface tension has a strong pulling force against a liquid having a small surface tension around it, and a surface tension gradient is generated; the liquid flows from a low surface tension to a high tension.

3. Question

Explain the figures 3a.

Answer

Thank you for your review. Fig. 3(a) is a figure in which the conductive patterns are placed under the conditions of the sintered power density of 13.6 J/cm2, 17.9 J/cm2, 20.3 J/cm2, 23.4 J/cm2, and sintered for 1 to 8 times, and the resistivity is recorded.

4. Question

Adhesion and pre-sintering temperature.

Answer

Thank you for your review. The manuscript shows that the adhesion of the conductive pattern is proportional to the pre-sintering temperature.

5. Question

Explain the figure 4.

Answer

Thank you for your review. Figure 4 shows the sintering of three samples at a pre-sintering temperature of 40 degrees Celsius, 60 degrees Celsius, and 80 degrees Celsius, and subjected to the same sintering parameters. The results show that the adhesion is proportional to the pre-sintering temperature. However, when the pre-sintering temperature is too high, the conductive pattern will be oxidized, resulting in a decrease in resistivity.